# AC Electroosmosis Effect on Microfluidic Heterogeneous Immunoassay Efficiency

**DOI:** 10.3390/mi11040342

**Published:** 2020-03-25

**Authors:** Marwa Selmi, Hafedh Belmabrouk

**Affiliations:** 1Department of Radiological Sciences and Medical Imaging, College of Applied Medical Sciences, Majmaah University, Majmaah 11952, Saudi Arabia; 2Laboratory of Electronics and Microelectronics, Faculty of Science of Monastir, University of Monastir, Environment Boulevard, Monastir 5019, Tunisia; ha.belmabrouk@mu.edu.sa; 3Department of Physics, College of Sciences at Zulfi, Majmaah University, Majmaah 11952, Saudi Arabia

**Keywords:** numerical simulation, immunosensor, alternating current electroosmosis (ACEO), detection time, performance biosensor

## Abstract

A heterogeneous immunoassay is an efficient biomedical test. It aims to detect the presence of an analyte or to measure its concentration. It has many applications, such as manipulating particles and separating cancer cells from blood. The enhanced performance of immunosensors comes down to capturing more antigens with greater efficiency by antibodies in a short time. In this work, we report an efficient investigation of the effects of alternating current (AC) electrokinetic forces such as AC electroosmosis (ACEO), which arise when the fluid absorbs energy from an applied electric field, on the kinetics of the antigen–antibody binding in a flow system. The force can produce swirling structures in the fluid and, thus, improve the transport of the analyte toward the reaction surface of the immunosensor device. A numerical simulation is adequate for this purpose and may provide valuable information. The convection–diffusion phenomenon is coupled with the first-order Langmuir model. The governing equations are solved using the finite element method (FEM). The impact of AC electroosmosis on the binding reaction kinetics, the fluid flow stream modification, the analyte concentration diffusion, and the detection time of the biosensor under AC electroosmosis are analyzed.

## 1. Introduction

Early detection of disease and successful remediation are desirable for effective healthcare. In order to detect the disease, there is great effort to improve the performance of existing devices and to design capable diagnostic devices such as biosensors. For example, a diagnostic device can provide specific and reliable detection of deadly diseases like cancer in the early stages [1]. There are several different types of biosensors used in the detection of diseases, namely, quartz crystal microbalances (QCMs), surface plasmon resonance (SPR) sensors, microcantilever beam-based biosensors, and immunosensors. Although the detection mechanisms are different, they are based on the interaction between two parts. The first part is a free target analyte (which may be an antigen or an antibody). The second part is an immobilized biological receptor on the sensitive layer. The role of the receptor (antibody or antigen) is to yield a quantified and measurable signal. A feature of this signal is related to the concentration of the target. Indeed, in a biological solution, the diluted analytes are continuously in motion. After injection, the analyte moves toward the interaction surface, where it associates or dissociates with the ligands. The analyte–ligand complex is the result of a combination of the mass transport phenomenon (e.g., convection and diffusion) and the molecule adsorption/desorption process on the reaction surface.

In the case of molecules with large association rate constants and high affinity, mass transport governs the reaction rate of the antigen–antibody [2]. The mass transport limitation will cause a diffusion boundary layer to develop near the reaction surface. The formation of this layer confines the chemical kinetics and globally reduces the performance of the biosensor. Several studies were published in the area of microfluidic biosensors [3,4,5,6]. This research describes the fundamental principle of heterogeneous immunoassay kinetics and the elaboration of numerical models based on the theoretical framework used for the design of biosensors. However, the problem remains treated a little more digitally than by experience. Lebedev et al. [7] and Mason et al. [8] theoretically studied the kinetic transport and modeling process of diffusion and convection phenomena in a biological sensor called BIACORE, to determine the effects of the limitations of the transport phenomenon. They studied the chemical kinetics of the association and dissociation processes by taking into account two reaction surface models: a surface at a single binding site and a surface at two binding sites. Sikavitsas et al. [9] developed a theoretical model to study the phenomenon of chemical species transport in a microfluidic optical biosensor. This model was used for the accurate prediction of kinetic parameters that helped to improve the performance of this biosensor. Lynn et al. [10] theoretically studied passive mixing structures to improve the mass transfer of the analyte. This study aims to increase the fluid velocity and, subsequently, the mass transport (by diffusion and by convection) at the surface of the biosensor.

Alternating current (AC) electrokinetic (ACEK) forces represent the most useful tool to manipulate mixing in microfluidic devices. This method is characterized by three sub-areas, known as AC electrothermal (ACET), AC electroosmosis (ACEO), and dielectrophoresis (DEP) forces [11,12,13]. DEP is applied, for instance, to control particles and detect cancer cells from the blood [14]. DEP is greatly used in biological applications for the separation of virus particles [15,16], the separation of nanoparticles [17], droplet manipulation [18], and trapping and manipulation of single cells and particles [19]. Dielectrophoresis (DEP) and alternating current electrothermal (ACET) flow are also combined to realize continuous particle trapping, switching, and sorting [20]. ACEO is used to move fluids with low conductivity. The frequency of the applied field is in general smaller than 100 kHz. This technique is irrelevant when the electrical conductivity of the biological fluid or the solution buffer is large [21]. 

There are only a few papers on the effect of AC electroosmosis on the transport mechanism and the binding reaction. A detailed study of the effect of this mechanism is needed to understand its contribution to the efficiency of heterogeneous immunosensors.

Heterogeneous immunoassays in biosensors take advantage of the interaction between a biological receptor ligand and free target analyte. In fact, the target analyte may be an antigen or an antibody, whereas the receptor molecule may be an antibody or, inversely, an antigen. The ligand is fixed on the reaction surface. The recognition of analytes may be exploited for disease detection. Some inefficient mass transport in the micro-system provides the development of a diffusion boundary layer, which has a direct impact on the binding reaction between ligand and analyte. This yields a limitation of the biosensor performances. An efficient immunosensor must accomplish high capture efficacy in the smallest time. Furthermore, the conditions to achieve more rapid transport and an optimum reaction are essential.

In order to enhance the reaction rate, several techniques were elaborated. Many of these techniques take advantage of the AC electrokinetic impact on improving the transport rating of reactants to a sensitive surface situated in the wall of a micro-channel. It is well known that AC electrokinetics may be categorized into three types of forces: electrothermal force, dielectrophoresis, and electroosmosis [4,12]. Recently, numerous technical and numerical surveys based on AC electrokinetics were carried out to enhance the performance of micro-fluidic biosensors. The AC electrothermal effect (ACET) was massively involved in the last decade of research on different biological technologies [22,23,24,25,26,27,28,29]. The ACET effect is effective at high frequencies (larger than 100 kHz) in a solution having a large electrical conductivity σ (greater than 0.002 S/m) [30]. Feldman et al. [21] performed experimental and numerical investigations. They used biotin/streptavidin as a heterogeneous assay, in which biotin is localized and fluorescently labeled streptavidin is suspended in a high-conductivity buffer (σ = 1 S/m). They showed that heterogeneous binding may be enhanced by electrothermal micro-stirring. When a 10 V_rms_ potential was applied, the reaction rate became nine times smaller.

Sigurdson et al. [4] studied electrothermal fluid flow. Both experimental and numerical analyses were carried out. This paper revealed that the binding rate of heterogeneous immunoassays was improved seven-fold when a value of 6 V_rms_ was applied.

Huang et al. [6,29] investigated two- and three-dimensional (2D and 3D) binding reaction kinetics. They used the finite element method. The reaction was achieved between two proteins, namely, C-reactive protein (CRP) and immunoglobulin G (IgG). The binding rate of heterogeneous immunoassays for CRP could be improved by a factor of two when a 15 V_rms_ voltage was applied. 

Selmi et al. also contributed to this area with some works summarized in our previously published articles [24,25,26,27].

ACET flow is present in a wide range of experimental and numerical studies in the field of biological applications. Nevertheless, there are limited works that studied the effect of AC electroosmosis on the transport and binding reaction. Understanding the relative contribution of this mechanism in heterogeneous immunosensors is required.

The present work aims to study the importance of the ACEO mechanism for the kinetics of the antigen–antibody binding reaction in a heterogeneous immunosensor of CRP antigen with its corresponding antibody anti-CRP. To accomplish this work, a set of equations were used to describe the problem, which was solved using the finite element method. The development of a mathematical model describing the ACEO effect is presented. In addition, the paper deals with the optimization of the biosensor design in order to investigate the impact of changing the geometry of the device on the binding. It also displays the distribution of the velocity, the concentration field, and the topology of the fluid under the ACEO effect.

## 2. Description and Formulation of the Problem 

Heterogeneous immunoassays are defined as a biosensor surface-based attachment between antigen and antibody. Either species may be a free target analyte in the sample volume, while the other is an immobilized biological receptor ligand in the solid phase. The target analyte diffuses within the microchannel and reacts with the ligand at the surface of the biosensor. For microfluidic-based sensors, which have dimensions on the order of micrometers, the mass transport is totally diffusive and convective. The flow is not turbulent. The reaction kinetics in the immunoassay biosensor are affected by the limitation of mass transport, which causes inordinately long detection times, leading to the limitation of the overall biosensor performance [31,32,33]. These issues resulted in increasing interest in the search for more efficient mass transport for microfluidic immunosensors.

In the present paper, we aimed to analyze the effect of the electroosmotic force on the kinetics of a specific binding reaction, namely, CRP–anti-CRP), in a microchannel configuration with integrated electrodes.

### 2.1. Immunoassay Geometry Model

The schematic of the simulation geometry is shown in Figure 1. It consisted of a cuboid microchannel having a reaction surface on the bottom wall. Two pairs of electrodes were placed on the top and bottom walls. Thus, an electric field was generated, and this led to the production of an electroosmotic force. A two-dimensional geometry was investigated in this paper. The length and width of the reaction surface were 20 µm and 1.5 µm, respectively, whereas the length and width of the microchannel were 250 µm and 40 µm, respectively. The electrodes had the same length, which was equal to 30 µm. The electrode thickness was assumed to be negligible. The height H of the channel was equal to 40 µm.

### 2.2. Electroosmotic Flow Modeling 

AC electroosmosis is classified as one aspect of AC electrokinetics. It is produced by the motion of ions along the surface of an electrode in an electric double layer via a tangential electric field. The electric double layer is a charged solution formed near the liquid–solid interface in response to the spontaneously formed surface charge, due to the contact of the flow with solid surfaces. When the electric field is applied, an electroosmotic flow is produced and moves the charged liquid. Then, a force is imposed on the positively charged solution in the vicinity of the wall. Thus, the motion of fluid in the direction of the electric field is initiated.

The fluid slip velocity on the electrode was deduced from the Helmholtz–Smoluchowski relationship, which links the electroosmotic velocity to the tangential component of the applied electric field [34,35,36,37,38].
(1)$$u_{ACEO}=\frac{\varepsilon_{0}\varepsilon_{r}\xi_{0}}{\mu }E_{t},$$
where $\varepsilon_{0}$ is the permittivity of free space (F/m), $\varepsilon_{r}$ represents the relative permittivity of the fluid (dimensionless), μ denotes the dynamic viscosity (Pa∙s), $\xi_{0}$ is the zeta potential at the inner wall, and $E_{t}$ is the tangential component of the applied electric field.

The flow velocity in the microchannel was described by the Navier–Stokes equations for laminar flows. The fluid was assumed to be incompressible and Newtonian. The continuity and motion equations can be written as
(2)𝛻.u=0,
(3)$$\rho \frac{\partial u}{\partial t}+\rho \left(u.𝛻\right)u=-𝛻p+\mu {𝛻}^{2}u+F.$$

The velocity vector is denoted by u, with 2D Cartesian components of (*u*, *v*); *p* is the pressure. The flow is presumed to be isothermal. The density of the fluid is ρ=1000 kg/m^3^ and its kinematic viscosity is $\nu =\frac{\mu }{\rho }=10^{-3}$ m^2^/s. The fluid density and its kinematic viscosity are supposed to be constant.

The only non-zero component of the inlet flow was the longitudinal component in the *x*-direction. It had a parabolic profile in the transverse *y*-direction, i.e., $u\left(0,y\right)=4v_{ave}\frac{y}{H}\left(1-\frac{y}{H}\right)$, where $v_{ave}=10^{-4}$ m/s is the average speed at the channel inlet. At the *y* channel outlet, we assumed that the fluid discharged in open air. At all solid walls, slip velocity boundary conditions were applied. Initially, the fluid was assumed to be at rest.

### 2.3. Electric Modeling

The Poisson equation was solved to compute the electrical potential V.
(4)$$\Delta V=0\;\mathrm{and}\;\vec{E}=-𝛻V.$$

Concerning the above electrostatic equation, the boundary conditions were of two kinds. At the electrodes, the electric potential was $V=\pm \;V_{max}$. Otherwise, the electric insulation condition was applied.

### 2.4. Analyte Concentration Modeling 

To compute the analyte concentration ${\left[A\right]}_{surf}$ at the sensitive surface, the analyte mass transport equation was used. Actually, a small quantity of a biological analyte was added to the fluid. The used analyte was C-reactive protein (CRP). A fraction of this analyte was transported by convection and diffusion toward the sensitive membrane. The analyte transport mass equation is given by
(5)$$\frac{\partial \left[A\right]}{\partial t}+u.𝛻\left[A\right]=D\Delta \left[A\right]+G,$$
where [A] is the analyte concentration.

The analyte diffusion coefficient D = 2.175 × 10^−11^ m^2^/s is supposed to be constant. The reaction rate G = 0, since no reaction occurs in the fluid bulk. The above equation contains a convective term and a diffusion term.

The boundary conditions related to the above equation were as follows: the inlet concentration was an adjustable constant $\left[A_{0}\right]$; at the outlet, the condition 𝛻[A]=0 was applied. At the reaction surface, a balance between the diffusive flux and the fluxes due to adsorption and desorption takes place. This balance is expressed by
(6)$$-D\frac{\partial \left[A\right]}{\partial t}=k_{on}{\left[A\right]}_{surf}\left\{{\left[B\right]}_{0}-\left[AB\right]\right\}-k_{off}\left[AB\right],$$
where $\left[B_{0}\right]$ is the concentration of free antibodies, and [AB] represents the complex concentration at the reaction surface. The rate constants related to the association and dissociation are $k_{on}$ and $k_{off}$, respectively.

The remaining portions of the walls were assumed to be impermeable, and no interaction with the analyte occurred at these parts. The initial value of the analyte concentration was [A](x,y,t=0)=0.

### 2.5. Immunoassay Surface Reaction Modeling 

The analyte was transported by diffusion and convection toward the sensitive membrane. Its concentration at the surface is designated by ${\left[A\right]}_{surf}$. Then, it reacted with the antibody ligand which was immobilized on this surface. Therefore, a complex AB was produced according to the following scheme:(7)$${\left[A\right]}_{surf}+\left[B\right]{⇌}_{k_{off}}^{k_{on}}\left[AB\right].$$

The antibodies B and the complex AB were immobilized on the surface. They were not transported away from this surface, whether by diffusion or by convection.

A first-order Langmuir adsorption model was used to represent the binding reaction between the analyte and the ligand [31]. The temporal evolution of the complex concentration [AB] is given by
(8)$$\frac{\partial \left[AB\right]}{\partial t}=k_{on}{\left[A\right]}_{\mathrm{surf}}\left\{\left[B_{0}\right]-\left[AB\right]\right\}-k_{off}\left[AB\right].$$

The association constant $k_{on}$ related to the CRP–anti-CRP binding interaction was 10^7^ M^−1^∙s^−1^, whereas the dissociation constant $k_{off}$ was 2.6 × 10^−2^∙s^−1^. The above equation required only an initial condition, which was given by [AB](t=0)=0.

### 2.6. Numerical Method 

In this work, the phenomenon of the transport of target analyte within the biosensor chamber was modeled by the transport mass equations, and the binding reaction kinetics at the sensing area was modeled by the transport equations based on the first-order Langmuir absorption model. The latter system of equations was built in a two-dimensional (2D) system and was solved using the finite element method (FEM) [39]. Firstly, the simulated domain was divided into triangular elements. The electrode surfaces and the sensing area were refined with a better mesh quality. Secondly, the computation was started by estimating the electric field, followed by the velocity field in order to obtain the electroosmotic flow effect on the binding reaction kinetics. Finally, we solved the convection–diffusion equations of analyte concentration coupled with the first-order Langmuir model of the diffusion equation of the ligand at the reaction surface. It should be noted that transport equations were time-dependent. The concentration of the complex produced at the reaction surface could be computed from the local concentration using numerical integration methods. The integration was performed over the entire reaction surface.

## 3. Results and Discussions 

### 3.1. Electroosmotic Flow 

During the immunoassay process, the analyte diffuses slowly from the bulk toward the reaction surface. Accordingly, the diffusion boundary layer starts to develop on the reaction surface; this hinders the binding reaction and restrains the response time and overall biosensor performance. In order to enhance the association and dissociation phase of the binding reaction, we applied an electric field to the flow, and an electroosmotic flow was generated. Furthermore, the electroosmotic mechanism induced a nearby force on the solution that was positively charged. This led to a flow in the direction of the electric field. The viscous transport in this direction was due to velocity gradients perpendicular to the wall.

The changes in the fluid topology provided by the electroosmotic flow led to an enhancement of the mass transport toward the reaction surface and substantially increased the analyte–ligand association and dissociation binding rate. 

Figure 2 shows the velocity field distribution in the case of the application and the absence the electric field. The average inlet velocity field applied at the inlet microchannel was fixed at $v_{ave}=0.1\;mm/s$ and the electric potential was 0.1 V. As mentioned above, a parabolic velocity profile was used at the channel inlet. As illustrated, the behavior of the fluid in the microchannel was considerably different in the case with the presence of an electric field compared with the case without an electric field. Figure 2a displays the velocity profile when the electric field was not applied; it is clear that the flow was laminar throughout the microchannel. The current lines were parallel to the longitudinal axis and the flow was fully developed. However, when the electric field was applied, we see that certain circular vortices appeared, which increased the mixing quality significantly (see Figure 2b), as shown in previous work [34]. This figure also gives us information that the vortices were close to the wall of the electrodes and depended on their positions. 

Figure 3 presents the analyte concentration distribution at t = 5 s in the case of an applied electric field and in its absence. The initial value of the analyte concentration at the inlet of the microchannel was 6.4 µmol/m^3^. As clearly shown in Figure 3a, for the distribution of the analyte concentration without the ACEO effect, the flow was parabolic and the replenishment of consumed analyte close to the binding surface was influenced only by the convection–diffusion mass transport. Therefore, the binding reaction between the target analyte and the immobilized ligand was restrained by the viscous effects near the surface reaction, which induced the development of the diffusion boundary layer and limited the immunoassay biosensor performance. However, when the voltage was turned on, the analyte concentration distribution was distorted by the electroosmotic flow (see Figure 3b). Thus, the ACEO produced a circular flow which caused fluid mixing. This yielded a microfluidic agitation which enhanced the transport of analytes toward the sensitive surface, increasing the chance of association and disassociation between antigen and antibody. Thus, the ACEO effect could considerably break the depletion zone compared to the static case, in which the sample solution was not reconstituted in the vicinity of the reaction surface. Therefore, the ACEO approach improved the mass transfer and enhanced the analyte–ligand binding. Thus, it enhanced the response of the biosensor.

### 3.2. Immunoassay Binding Reaction Kinetics

In order to assess the ACEO effect on the binding reaction kinetics of the CRP, we numerically simulated the temporal evolution of the association and dissociation of the analyte–ligand complex at the reaction surface. Figure 4 exhibits the temporal evolution of the analyte–ligand complex concentration with or without applying external voltage. The complex concentration increased steeply over a short time. This means that the association mechanism was dominant. Then, a balance between the association and dissociation mechanisms occurred, and the saturation regime was obtained. At t=1500 s, the injection of analyte was stopped, and the dissociation phase began. This led to an increase in the complex concentration.

As illustrated in Figure 4, the binding rates of both association and dissociation phases were determined for the two investigated configurations (i.e., with and without the ACEO effect). In the case when the electric field was applied, a strong binding enhancement was observed compared to the case without an applied voltage. This was the key reason for the ACEO efficiency in improving the biosensor immunoassay. Indeed, the electroosmotic flow generated vortices, enhancing the flow mixing, which improved the mass transport of the analyte from the bulk of the microchannel toward the reaction surface. Subsequently, the analyte depletion zone caused by diffusion limitation could be effectively overcome, which resulted in enhancement.

Now, we define the detection time as the response time of the biosensor, i.e., the time in which the complex concentration at the reaction surface achieved 95% of its saturation value. This parameter was used to quantify the ACEO effect on the binding kinetics reaction. Table 1 lists the response time in both cases studied in this work. As found, the detection time was smaller when the electric field was applied. The ACEO approach could improve the binding reaction analyte–ligand interactions. Therefore, the biosensor performance was boosted.

To investigate the dependence of the applied voltage on the binding reaction, we performed a numerical simulation on the evolution of the complex concentration by varying the applied voltage. As shown in Figure 5, an increase in binding rate was obtained as V increased from 0.1 to 0.4 V. As found, the enhanced CRP binding was proportional to the increase in voltage. Indeed, the response time was reduced by increasing the voltage (Table 2), and we also noted that the percentage of the reduction in time started slowing down from the tension at V = 0.3 V, which signified there existence of an optimum ACEO.

## 4. Conclusions

This study focused on illuminating the ACEO effect on the binding reaction kinetics in a heterogeneous immunoassay in the microchannel of a biosensor. Therefore, a 2D numerical simulation was performed using the finite element method. By applying an electric field through the electrodes, an electroosmotic flow was generated within the flow, which significantly improved the mixing flow and led to enhanced mass transport of the analyte from the bulk toward the sensing area. Therefore, the ACEO hindered the growth of the limited boundary layer and could remove the depletion zone, whereby the analyte concentration was not reconstituted on the surface of the sensor. In addition, the voltage increase may considerably reduce the response time. Therefore, the ACEO can enhance biosensor performance.

Future work may be oriented in different directions. Studies related to the investigation of the confinement effect or the development of a new adsorption model can be carried out. This study may also be supplemented with an experimental investigation.

## Figures and Tables

**Figure 1 micromachines-11-00342-f001:**
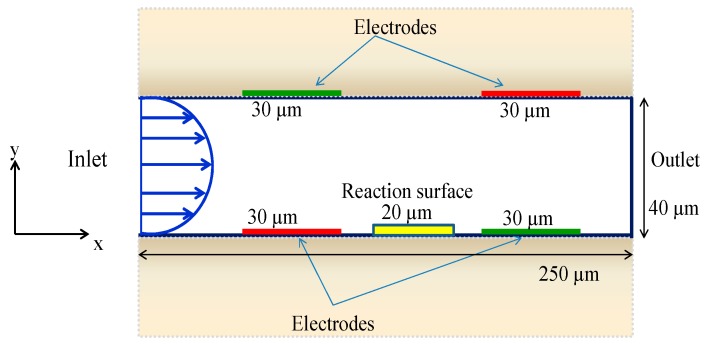
Schematic model geometry of the microchannel of the biosensor.

**Figure 2 micromachines-11-00342-f002:**
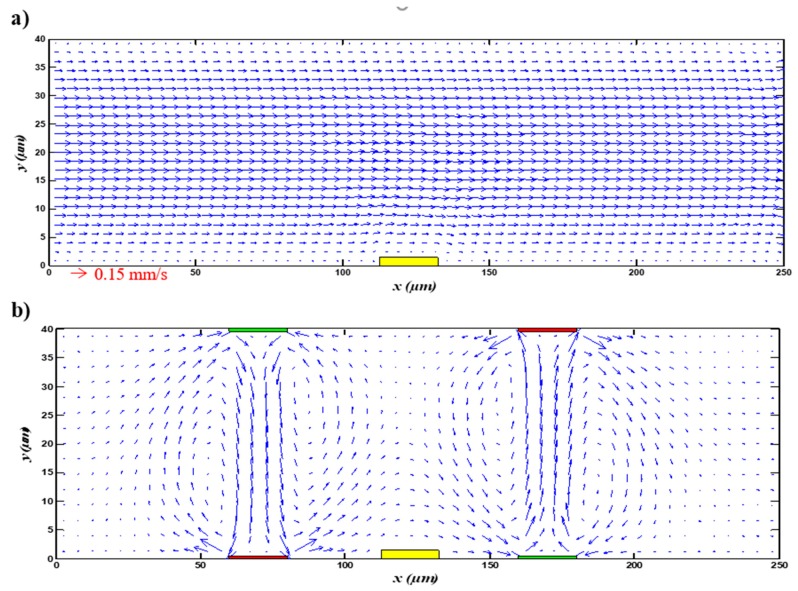
Velocity field (**a**) in the absence of an electric field and (**b**) in the presence of an applied electric field.

**Figure 3 micromachines-11-00342-f003:**
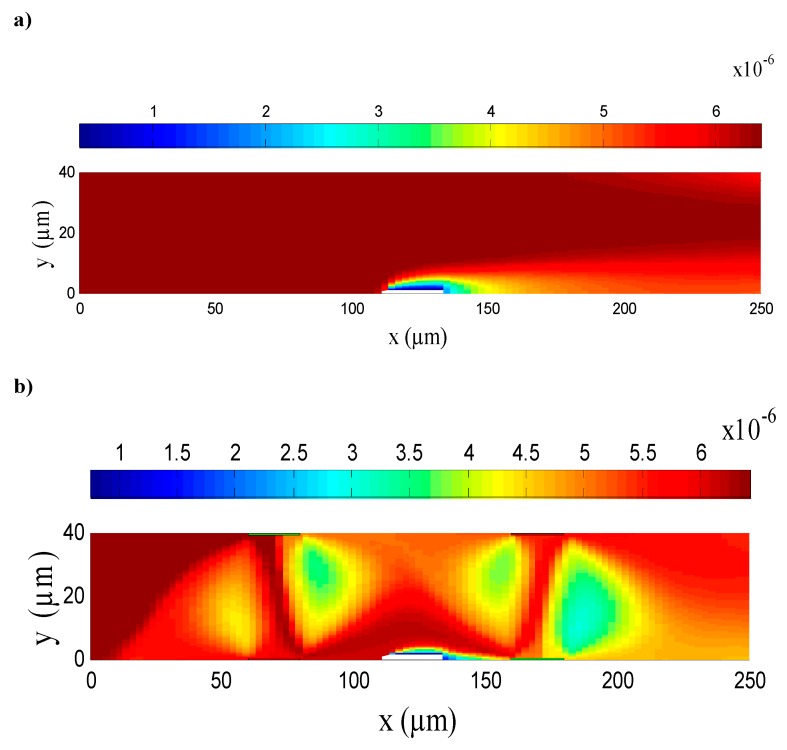
Analyte concentration distribution (**a**) in the absence of an electric field and (**b**) in the presence of an applied electric field.

**Figure 4 micromachines-11-00342-f004:**
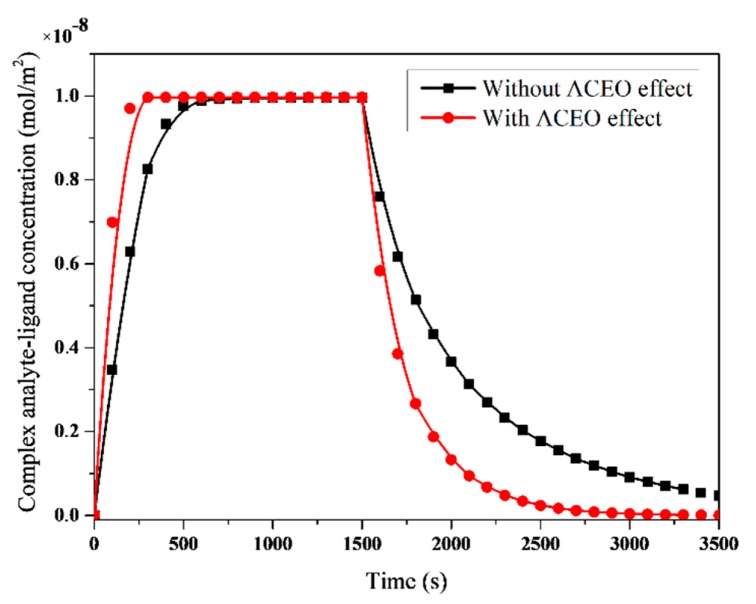
Temporal evolution of the analyte–ligand complex concentration with or without applying voltage.

**Figure 5 micromachines-11-00342-f005:**
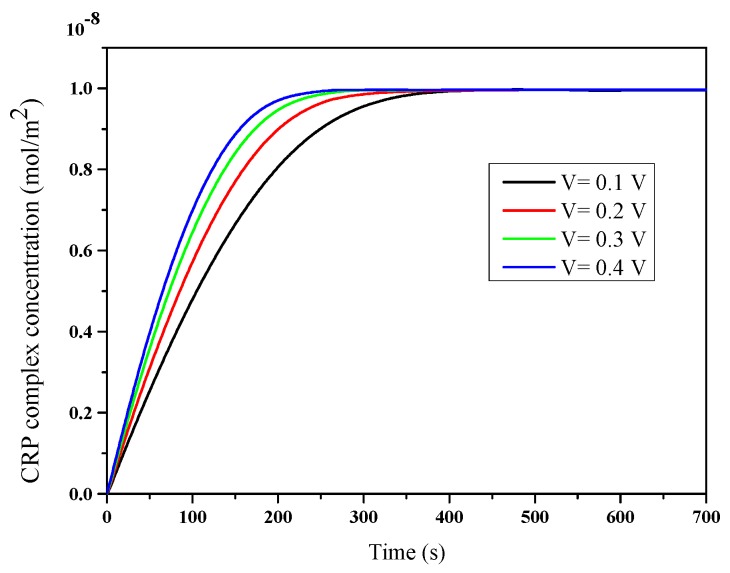
Effect of various voltages on the C-reactive protein (CRP) binding reaction.

**Table 1 micromachines-11-00342-t001:** Response time for C-reactive protein (CRP) binding reaction. ACEO—alternating current electroosmosis.

Case	Detection Time (s)
Without ACEO effect	417
With ACEO effect	180

**Table 2 micromachines-11-00342-t002:** Response time for C-reactive protein (CRP) binding reaction and percentage reduction.

Applied voltage (V)	Detection Time (s)	Reduction Percentage (%)
0.1	291	30.21
0.2	232	44.36
0.3	200	52.03
0.4	180	56.83

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
