# Peer review of "AC Electroosmosis Effect on Microfluidic Heterogeneous Immunoassay Efficiency"

_micromachines, 2020, doi:10.3390/mi11040342_

Round 1

Reviewer 1 Report

file attached

Author Response

Response to Reviewer 1 Comments

The article entitled “AC electroosmosis effect on microfluidic heterogeneous immunoassays efficiency” by M. Selmi and H. Belmabrouk shows mixing and reaction enhancement effects in a microchannel exploiting the electrokinetic phenomena. Since the structure of the article is poor and difficult to understand the authors’ claim, I do not recommend to publish this article in Micromachines. Detailed comments are provided below.

Point 1:

  1. Sentences

The structure is very poor: some sentences (p. 3, lines 120–132, p.6 lines 217–235) just repeat what has been already mentioned. On the other hand, some symbols (e.g., ε0, εr) are used without definition or with definition but not given at the first appearance. Furthermore, English should be entirely revised. In addition, reference should be correctly formatted

Response 1:

  1. Some sentences have been removed to avoid the repetition.
  2. The definitions of all symbols (e.g., ε0, εr, , ,…) have been added in the new version manuscript. Definitions are given at the first appearance of each symbol.
  3. English have been revised.
  4. References are automatically formatted using EndNote.

Point 2:

  1. Electrodes

The authors should explain the design of the electrodes (why on both top and bottom walls? For instance).

Response 2:

It is well known that the mixing is as very slow in particle diffusion processes, such as in micro-biosensor. Therefore, an efficient mechanism for mixing is desired in order to enhance the binding reaction kinetics. Thus, the application of an external electric field is widely used in flow mixing. According to Amir Shamloo et al. [34], a geometric model with four symmetric electrodes on the wall of the mixing chamber is highly recommended to use as an optimum case, which greatly contributed to a remarkable mixing quality (about 99.4%).

Point 3:

  1. What is the value of vave? Is vave supplied for the case of Fig. 2a? Show velocity / concentration scale in Fig. 2 and Fig. 3, respectively.

Response 3:

  1. The value of has been added.  

  1. Concerning the question “Is vave supplied for the case of Fig. 2a”, a paragraph related to Fig 2 has been modified as follows:

The average inlet velocity field applied at the inlet microchannel is fixed at  and the electric potential is 0.1 V. As above mentioned, a parabolic velocity profile is used at the channel inlet.

  1. 2 and Fig. 3 have been redrawn. The scales have been added in the new version.

Point 4:

  1. Reaction condition

If the analyte concentration is high, the device would not need the external force (the detection would be completed in few minutes). What is important may be the enhancement beyond the detection limit. What will be happen for that case? The applied voltage often used in literature is about one order higher than the present study. Why did the authors choose O(10-1) V?

Response 4: Please provide your response for Point 2. (in red)

Medical analysis laboratories is widely used the micro devices in diagnosis and treatment of various diseases for individuals. A problem that arises in these lab-on-chip devices concerns the transport of the liquid samples in the chip, which are of very small dimensions. In addition, the physician can certainly require many experiences and must use small sample every time. An alternative method for transporting fluid in the samples is through electrokinetic effects, such as the ACEO, where charged ions in the solutions are subjected to an electric field. The effect is greatly used to enhance the mass transport in microchannel.

The ACEO produces a circular flow which causes fluid mixing. This yields a microfluidic agitation which enhances the transport of analytes towards the sensitive surface providing more chance for association and disassociation between antigen and antibody.

The ACEO is produced when an electric filed is applied even when a low voltage. Some researchers have devoted their work to this topic;Qingming Hu et al. revealed the improving microfluidic heterogeneous immunoassay using the electroosmosis with a voltage variation of (2V, 4 V, 6 V, 8 V, 16 V, 32 V). Amir Shamloo et al. demonstrated the flow high mixing using 0.05, 0.1, 0.2, 0.5 V. Yujing Song el al. also has proved a significant enhancement of ACEO using as voltage  V0 from 1 to 4 V.

Point 5:

  1. Fig. 4 Why the concentration suddenly decreases at t ~ 2000 s? and Why the decreasing rate is different for w/o and w/ ACEO?

Response 5:

The binding reaction in a biosensor immunoassay is formed by two phases; the association phase (  and the dissociation phase ( . Indeed, Fig. 4 shows the temporal evolution of the complex concentration at the reaction surface of the biosensor with and without an applied electric field. In both cases, the complex concentration increases steeply over a short time which presents the association phase and then it reaches a saturation value. At t = 1500 s, the injection of analyte is stopped. We observe the complex concentration decreases due to the dissociation process. When the electric field is applied, the association and dissociation increases and declines rapidly than in the case without an applied electric filed. The main reason is the effect of the ACEO on the binding reaction.

The following sentence has been added to the new version

“Figure 4 exhibits the temporal evolution of the complex analyte-ligand concentration with or without applying external voltage. The complex concentration increases steeply over a short time. This means that the association mechanism is dominant. Then a balance between the association and dissociation mechanisms occurs and the saturation regime is obtained. At t=1500 s, the injection of analyte is stopped and the dissociation phase starts. This leads to the increase of the complex concentration.”

Reviewer 2 Report

The manuscript presents a numerical investigation of the AC electroosmosis on the kinetics binding reaction, fluid flow stream modification, analyte concentration diffusion, and detection time of biosensor. Physical and mathematical models based on a microchannel were designed for the induction of velocity field and analyte concentration distribution under the external applied electric field. Numerical simulation was conducted based on the design. The result showed ACEO enhances biosensor performance. The manuscript is overall written clearly. I can recommend this paper for publication if the authors can address the following comments.

 (1) There are quite a few spelling mistakes and sometimes sentences do not make sense. Please review the paper and improve English.

(2) Why is AC used instead of DC electric field? The DC electric field drives both fluid electroosmosis and particle electrophoresis, the authors should discuss the different or consistent effects for the electroosmosis under AC electric field compared with DC electric field.

(3) Fig.2 is obscure, increase the front and redraw it. Moreover, it is given as x(μm), y(μm) in Fig.2, which reads like 4×10-5 μm. It is incorrect and revises it to keep consistent with that in Fig.1.

(4) Why not carrying out the experiment to verify the simulation?

(5) Critical commentary is needed from the author who eventually recommends directions for further research.

(6) In Line 67, this work has cited lots of valuable publications, however, some recent related references applying DEP for particle manipulation and biological sorting should be cited as:

"Continuous flow separation of particles with insulator-based dielectrophoresis chromatography." Analytical and Bioanalytical Chemistry (2020): 1-12.

"Continuous particle trapping, switching, and sorting utilizing a combination of dielectrophoresis and alternating current electrothermal flow." Analytical Chemistry 91.9 (2019): 5729-5738.

"Analysis of bacteriophages with insulator-based dielectrophoresis." Micromachines 10.7 (2019): 450.

"Tunable droplet manipulation and characterization by AC-DEP." ACS applied materials & interfaces 10.42 (2018): 36572-36581.

"High-throughput separation, trapping, and manipulation of single cells and particles by combined dielectrophoresis at a bipolar electrode array." Analytical Chemistry 90.19 (2018): 11461-11469.

"Continuous separation of nanoparticles by type via localized DC-dielectrophoresis using asymmetric nano-orifice in pressure-driven flow." Sensors and Actuators B: Chemical 250 (2017): 274-284.

"Separation of nanoparticles by a nano-orifice based DC-dielectrophoresis method in a pressure-driven flow." Nanoscale 8.45 (2016): 18945-18955.

Author Response

Response to Reviewer 2 Comments

The manuscript presents a numerical investigation of the AC electroosmosis on the kinetics binding reaction, fluid flow stream modification, analyte concentration diffusion, and detection time of biosensor. Physical and mathematical models based on a microchannel were designed for the induction of velocity field and analyte concentration distribution under the external applied electric field. Numerical simulation was conducted based on the design. The result showed ACEO enhances biosensor performance. The manuscript is overall written clearly. I can recommend this paper for publication if the authors can address the following comments

Point 1: There are quite a few spelling mistakes and sometimes sentences do not make sense. Please review the paper and improve English.

Response 1:

The manuscript has been reviewed to improve English and avoid spelling mistakes.

Point 2: Why is AC used instead of DC electric field? The DC electric field drives both fluid electroosmosis and particle electrophoresis, the authors should discuss the different or consistent effects for the electroosmosis under AC electric field compared with DC electric field.

Response 2: Please provide your response for Point 2. (in red)

The AC electrokinetic phenomena includes three broad areas namely; the dielectrophoresis (DEP), electrothermal forces and AC electroosmosis. DC electrokinetics has been widely used for lab-on-a-chip applications such as electroosmotic pumping and capillary gel electrophoresis for DNA fractionation. In contrast, AC electrokinetics has received significantly less attention. AC electrokinetics have the advantages of largely avoiding electrolysis, and operating at lower voltages (1–20 V), which is important for portable systems.

The phenomenon of electroosmosis is based on the formation of electric double layer. Direct current electroosmosis (DCEO) has the disadvantages of high electrochemical reactions and electrolysis at the electrode surfaces, as well as the formation of bubbles and microscale pH gradient due to the high voltage (kilovolts) usage.

To overcome these issues, ACEO has been proposed, which has the advantages of operating at low voltages and generating non-uniform electric field.

According to Alinaghi Salaria, Michael Thompson (2018), the thickness of double layer is inversely proportional to the electrolyte conductivity meaning that for high electrical conductivity fluids (e.g. most biological fluids), the thickness of double layer is small and therefore, the resultant ACEO effect becomes weak. Thus, ACEO is more efficient if applied to liquids with low electric conductivities (  S/m).

Point 3: Fig.2 is obscure, increase the front and redraw it. Moreover, it is given as x(μm), y(μm) in Fig.2, which reads like 4×10-5 μm. It is incorrect and revises it to keep consistent with that in Fig.1.

Response 3:

The Fig. 2 has been redrawn in the new version.

Point 4: Why not carrying out the experiment to verify the simulation?

Response 4:

In the next future work, we intend to extend this study and accomplish the experimental part. This issue has been mentioned in the conclusion.

Point 5: Critical commentary is needed from the author who eventually recommends directions for further research.

Response 5:

The following sentence has been added in the conclusion:

“Future works may be oriented in different directions. They may be related to the investigation of the confinement effect or the development of a new adsorption model. This study may also be completed by an experimental part.”

Point 6: In Line 67, this work has cited lots of valuable publications, however, some recent related references applying DEP for particle manipulation and biological sorting should be cited as:

"Continuous flow separation of particles with insulator-based dielectrophoresis chromatography." Analytical and Bioanalytical Chemistry (2020): 1-12.

"Continuous particle trapping, switching, and sorting utilizing a combination of dielectrophoresis and alternating current electrothermal flow." Analytical Chemistry 91.9 (2019): 5729-5738.

"Analysis of bacteriophages with insulator-based dielectrophoresis." Micromachines 10.7 (2019): 450.

"Tunable droplet manipulation and characterization by AC-DEP." ACS applied materials & interfaces 10.42 (2018): 36572-36581.

"High-throughput separation, trapping, and manipulation of single cells and particles by combined dielectrophoresis at a bipolar electrode array." Analytical Chemistry 90.19 (2018): 11461-11469.

"Continuous separation of nanoparticles by type via localized DC-dielectrophoresis using asymmetric nano-orifice in pressure-driven flow." Sensors and Actuators B: Chemical 250 (2017): 274-284.

"Separation of nanoparticles by a nano-orifice based DC-dielectrophoresis method in a pressure-driven flow." Nanoscale 8.45 (2016): 18945-18955.

Response 6:

The following paragraph has been added (lines 67-71). Some new references have also been added:

“DEP is greatly used in biological applications for separation virus, particle, [15, 16], separation nanoparticles [17], droplets manipulation [18], trapping and manipulation of single cells and particles [19]. Dielectrophoresis (DEP) and alternating current electrothermal (ACET) flow are also combined to realize continuous particle trapping, switching, and sorting [20].”

Round 2

Reviewer 1 Report

Although I don't agree with the idea that only reduces the detection time few minutes and increases both the fabrication cost and process, I would say okay if the editor think it deserves. But my opinion is that the study just performed simulation with a design that someone said optimal. If it shows the proposed method opens new possibility that former chips cannot achieve, then it is worth to publish, of course.